# Study of Modifications Induced by Continued Direct Oral Anticoagulant Therapy during Atrial Fibrillation Ablation Procedures on Standard Hemostasis Parameters

**DOI:** 10.3390/jcm12062236

**Published:** 2023-03-14

**Authors:** Marie Muller, Julien Godet, Xavier Delabranche, Laurent Sattler, David Millard, Halim Marzak, Paul Michel Mertes, Annick Steib, Lelia Grunebaum, Laurence Jesel, Charles Ambroise Tacquard

**Affiliations:** 1Department of Anesthesia and Intensive Care, Hôpitaux Universitaires de Strasbourg, 67000 Strasbourg, France; 2Groupe Méthodes en Recherche Clinique (GMRC), Hôpitaux Universitaires de Strasbourg, Hôpital Civil, 67000 Strasbourg, France; 3Laboratoire d’Hématologie, Unité Hémostase, Hôpitaux Universitaires de Strasbourg, 67000 Strasbourg, France; 4Service de Cardiologie, Hôpitaux Universitaires de Strasbourg, 67000 Strasbourg, France; 5UMR 1260 INSERM Nanomedecine Regenerative, CRBS, Strasbourg University, 67200 Strasbourg, France

**Keywords:** atrial fibrillation ablation, heparin, ACT, direct oral anticoagulant, anti-Xa

## Abstract

Background: Unfractionated heparin (UFH) is used as an anticoagulant during the atrial fibrillation (AF) ablation procedure to prevent the occurrence of thromboembolic events. Guidelines recommend an activated clotting time (ACT) greater than 300 s (s) based on studies of patients treated with vitamin K antagonist (VKA) for their AF. However, direct oral anticoagulants (DOACs) have supplanted VKAs in AF and are now used as first-line therapy. It is recommended not to interrupt them during the procedure, which could interfere with the ACT measures. Objective: To assess the real-life relationship between ACT, DOAC concentrations, and UFH anti-Xa activity in patients treated by uninterrupted DOAC therapy. Methods: We conducted a single-center retrospective study. We analyzed consecutive patients with AF who underwent catheter ablation under DOAC therapy. Results: In total, 40 patients were included, including 15 (37.5%), 20 (50.0%), and 5 (12.5%) on rivaroxaban, apixaban, and dabigatran, respectively. Baseline ACT was significantly lower in the apixaban group. ACT was linearly correlated with the residual concentration of apixaban and dabigatran but not with rivaroxaban. After UFH injection, ACT was linearly correlated with the anti-Xa activity, regardless of DOAC. Patients in the apixaban group received a higher total dose of UFH during the procedure to achieve a target ACT > 300 s, which resulted in significantly higher anti-Xa activity during the procedure. Conclusion: Our results raise the question of optimal management of intra-procedural heparin therapy and highlight the limitations of the ACT test, particularly in patients on apixaban.

## 1. Introduction

Atrial fibrillation (AF) is the most common cardiac rhythm disorder, and its incidence is constantly increasing [1,2]. Catheter ablation has become the gold standard in the treatment of AF [3]. In addition to the inherent bleeding risk of the procedure, thromboembolic complications are a concern because of the presence of catheters in the heart chambers that can induce blood clot formation. Careful management of periprocedural anticoagulation is therefore essential. The current recommendation of the European Society of Cardiology states that oral anticoagulation is required at least one month before and two months after the ablation [4]. Additional anticoagulation with unfractionated heparin (UFH) is required during the procedure. The combination of an oral anticoagulant and heparin may increase the risk of bleeding. The activated clotting time (ACT) is used to monitor the effectiveness of intraprocedural anticoagulation. Data on anticoagulation management and ACT monitoring during the procedure have been obtained in patients treated with vitamin K antagonists (VKAs) [5,6]. Since 2009, direct oral anticoagulants (DOACs) have been increasingly used as first-line therapy for AF. However, guidelines for ACT goal and heparin management during AF ablation have been directly extrapolated from VKAs to DOAC patients. Several studies have shown that uninterrupted DOAC during AF ablation was a safe and efficient strategy. When patients receive DOACs, higher doses of UFH seem to be required to achieve an ACT > 300 as recommended, with differences between DOAC molecules [7,8,9,10]. The reliability of ACT to manage optimal anticoagulation during AF ablation is therefore questionable. The baseline anticoagulation state when DOAC is continued until the day of ablation is not reproductible between patients, and ACT is disturbed by the presence of DOAC and is not correlated to the DOAC concentration [11,12]. In view of these considerations, the present study aims to clarify the impact of uninterrupted direct oral anticoagulant therapy during atrial fibrillation ablation procedures on standard hemostasis parameters.

## 2. Materials and Methods

### 2.1. Study Design and Procedure Protocol

We conducted a retrospective, single-center, observational study at the University Hospital of Strasbourg. The protocol was approved by the ethics committee of the Strasbourg medical school (CE-2020-25).

Patients referred in our center for AF catheter ablation between February 2019 and June 2021 were included. Direct oral anticoagulants (DOACs—rivaroxaban, apixaban, or dabigatran) were uninterrupted until the day of the AF ablation procedure, and hemostasis exploration before and during the procedure was evaluated. The last dose of DOAC was taken the evening before the procedure for rivaroxaban and the morning of the procedure for apixaban and dabigatran. Patients were included if they continued DOAC therapy until the procedure. They were excluded if they stopped their treatment before the procedure or if they were on VKA. Only patients scheduled first in the program were included to allow for extemporaneous hemostasis analysis by the laboratory and to guarantee a homogeneous population in relation with the pharmacokinetic of the DOAC.

This procedure was routinely performed under general anesthesia in the interventional cardiology unit.

The standard of care protocol for all patients with DOAC therapy maintained until the AF ablation procedure in our unit included the following blood sampling at the beginning of the procedure for specific anti-Xa/IIa activity, calibrated anti-Xa activity for UFH [13], and ACT. After heparin injection, ACT and UFH anti-Xa activity were assessed every 20 min until the end of the procedure. The first bolus of unfractionated heparin was injected immediately after transseptal puncture (100 IU.kg^−1^) according to HRS/EHRA/ECAS guidelines [13]. During the procedure, additional boluses of UFH (50 IU.kg^−1^) were administered to achieve an ACT greater than 300 s, if necessary. Transesophageal echocardiography was routinely performed to search for intracavitary thrombus and to guide trans-septal puncture. 

All blood tests were performed by the hemostasis laboratory. ACT was measured using a Hemochron Signature Elite system (Werfen). Anti-Xa activity (UHF and rivaroxaban, apixaban) was measured by a chromogenic anti-Xa assay (STA-Liquid anti-Xa, Stago, Asnières sur Seine, France) with a specific calibration for each molecule (STA-R Max, Stago, Asnières sur Seine, France). Anti-IIa activity (dabigatran) was measured by a chronometric assay using thrombin time (Hemoclot Thrombin Inhibitors, Hyphen Biomed) with a specific calibrator (STA-R Max, Stago, Asnières sur Seine, France).

### 2.2. Data Collection

For each patient, data collected from the medical record included: age, body mass index (BMI), type and dose of oral anticoagulant, time of last DOAC intake, type of AF, CHA2DS2-VAsc score, Cockcroft–Gault renal function, comedication such as antiplatelet agents, and postoperative complications.

### 2.3. Statistical Analysis

The statistical analysis of the anonymized data was carried out using R software version 4.2.0 (R Foundation for Statistical Computing, Vienna, Austria). Qualitative data were expressed as frequencies and proportions. Quantitative variables were described by their median and interquartile range (IQR). Linear associations were determined using Bayesian generalized linear models (glm) (rstanarm package). Posterior distribution summaries were retrieved (MCMC output including mean, median, quantiles, Gelman–Rubin convergence statistic, number of effective samples) using MCMCvis package. Posterior probabilities higher than 0.975 or lower than 0.025 were used as a threshold to determine differences between groups or to demonstrate linear associations between quantitative variables. 

The association between two variables was quantified by the posterior probability distribution of the (ß1) coefficient of the linear regression model (ß1 = 0 corresponds to an absence of linear association).

## 3. Results

### 3.1. Population and Procedural Characteristics

Forty patients referred for AF catheter ablation were included. The patients’ characteristics are described in Table 1. Only one (2.5%) patient was treated with both DOAC and clopidogrel. The times since the last DOAC intake were 4 (3–5), 18 (15–21), and 5 (4–7) hours before the procedure for apixaban, rivaroxaban, and dabigatran, respectively.

### 3.2. Baseline Anticoagulation State

The anticoagulation level before the first bolus of unfractionated heparin (UFH) is shown in Figure 1. The specific anti-Xa activity for rivaroxaban ranged from 25 to 334 ng.mL^−1^ with two (14%) patients under the therapeutic threshold (<30 ng.mL^−1^) and one (7%) with a supratherapeutic value (>300 ng.mL^−1^) (Figure 1A). In the apixaban group, the anti-Xa activity for apixaban ranged from 22 to 352 ng.mL^−1^, with one (5%) patient under the therapeutic threshold and three (15%) patients with a supratherapeutic value (Figure 1B). The anti-IIa activity for dabigatran ranged from 30 to 351 ng.mL^−1^ (Figure 1C). Baseline ACTs were at 157 (141–176), 126 (114–140), and 149 (133–189) seconds for rivaroxaban, apixaban, and dabigatran, respectively (Prob (dabi > api) = 0.99, Prob (riva > api) = 0.99, Prob (dabi > riva) = 0.55). Anti-Xa (UFH) activities were 1.6 (1.1–2.0), 1.8 (1.8–2.0), and 0.0 (0.0–0.0) UI.mL^−1^ for rivaroxaban, apixaban, and dabigatran, respectively. Baseline ACT was linearly associated with the residual concentration of DOAC for apixaban (mean value of the linear regression coefficient b_1_ = 0.147, Prob (ß1 > 0) = 0.99) and dabigatran (b_1_ = 0.334, Prob (ß1 > 0) = 0.99) but not for rivaroxaban (b_1_ = 0.05, Prob (ß1 > 0) = 0.75).

### 3.3. Effect of the Initial Bolus of UFH

The dose of the initial bolus of UFH was similar between groups at 97 (94–102), 97 (87–100), and 100 (94–103) UI.kg^−1^ for apixaban, rivaroxaban, and dabigatran, respectively (Prob (dabi > api) = 0.59, Prob (riva > api) = 0.09, Prob (dabi > riva) = 0.12).

The ACT increased by 126 (104–147), 120 (93–147), and 170 (124–213) seconds for apixaban, rivaroxaban, and dabigatran, respectively, after the initial bolus of UFH. Median anti-Xa activity after UFH bolus increased by 0.73 (0.55–0.91), 0.59 (0.38–0.81), and 1.88 (1.50–2.22) IU.mL^−1^ for apixaban, rivaroxaban, and dabigatran, respectively. ACT tended to increase more and anti-Xa activity increased significantly more after the initial bolus of UFH in patients treated with dabigatran than in those treated with rivaroxaban or apixaban (ACT: Prob (dabi > riva) = 0.97; Prob (dabi > api) = 0.96; Prob (riva > api) = 0.37; anti-Xa: Prob (dabi > api) = 0.99; Prob (dabi > riva) = 0.99; Prob (riva > api) = 0.16) (Figure 2A,B).

### 3.4. Anticoagulation during the Procedure

The median duration of the procedure was 80 (65–120) minutes in the rivaroxaban group, 100 (86–120) minutes in the apixaban group, and 95 (50–105) minutes in the dabigatran group. Procedure duration was not statistically different between DOACs (Prob (dabi > api) = 0.18; Prob (riva > api) = 0.08; Prob (dabi > riva) = 0.48). The total dose of UFH administered during the procedure to achieve and maintain an ACT > 300 s was 146 (129–195) UI, 115 (100–144) UI, and 145 (95–153) UI.kg^−1^ in the apixaban, rivaroxaban, and dabigatran group, respectively. Patients treated with apixaban received significantly more UFH than patients treated with rivaroxaban (Prob (api > riva) = 0.99), while there was no significant difference between patients treated with apixaban and dabigatran (Prob (api > dabi) = 0.90).

Maximal ACTs during the procedure were 322 (294–335), 330 (311–355), and 357 (349–363) seconds in patients treated with apixaban, rivaroxaban, and dabigatran, respectively. The maximal ACT tended to be higher in patients treated with dabigatran vs. apixaban (Prob (dabi > api) = 0.92) and was not different between rivaroxaban and apixaban (Prob (riva > api) = 0.89) or between dabigatran and rivaroxaban (Prob (dabi > riva) = 0.71) (Figure 3).

The maximal anti-Xa activity during the procedure was 2.8 (2.6–3.0), 2.2 (2.0–2.4), and 2.0 (1.8–2.0) UI.mL^−1^ in patients treated with apixaban, rivaroxaban, and dabigatran, respectively. Maximal anti-Xa activity was significantly higher in patients treated with apixaban vs. dabigatran or rivaroxaban (Prob (api > dabi) = 0.99, Prob (api > riva) = 0.99, Prob (riva > dabi) = 0.90).

### 3.5. Correlation between ACT and Anti-Xa UFH during the Procedure (after the Initial Bolus of UFH)

During the procedure, the increase in ACT correlated with the increase in UFH anti-Xa activity. For a 100-second increase in ACT, UFH anti-Xa activity significantly increased by 0.49 (0.40–0.59), 0.36 (0.27–0.43), and 0.89 (0.66–1.11) IU.mL^−1^ in patients treated with apixaban, rivaroxaban, and dabigatran, respectively (Prob (dabi > api) = 0.99; Prob (api > riva) = 0.68; Prob (dabi > riva = 0.99).

### 3.6. Complications after the Procedure

One patient developed a bleeding complication after the ablation procedure. This patient was treated with apixaban and developed bleeding from the soft palate after the procedure that required general anesthesia for endoscopic exploration. The residual apixaban concentration was 116.3 ng.mL^−1^. The total dose of UFH administered during the procedure was 15,000 IU corresponding to 221 IU.kg^−1^. The maximum anti-Xa activity of UFH and ACT during the procedure was 2.74 IU.mL^−1^ and 357 s, respectively. No thromboembolic event was recorded.

## 4. Discussion

This study explores the influence of uninterrupted DOAC therapy on hemostasis during AF catheter ablation.

### Main Findings

Our results showed the following: (i) baseline anticoagulant status was highly variable before the start of the procedure, with some patients above the therapeutic threshold and others below; (ii) ACT before UFH injection was linearly correlated with apixaban and dabigatran concentration but not with rivaroxaban concentration, and baseline ACT was significantly lower in patients treated with apixaban; (iii) during the procedure, ACT was linearly correlated with UFH anti-Xa activity, but ACT increased slower than anti-Xa activity in patients treated with apixaban and rivaroxaban when compared with dabigatran; and (iv) the total dose of UFH delivered during the procedure to achieve and maintain an ACT target over 300 s was higher in the apixaban group, resulting in a higher anti-Xa activity in this group, indicating that ACT monitoring may lead to an excessive use of UFH in this group.

The baseline anticoagulation level of DOACs was variable in our study, regardless of the molecule used, with some patients having infra- or supratherapeutic anticoagulation levels. DOACs are usually described as having a limited interindividual variability by the manufacturers, especially for rivaroxaban and apixaban, but some specific situations, such as severe hepatic or renal dysfunction, are known to be related to interindividual variability [14,15,16,17,18,19]. None of our patients had severe chronic renal failure that could explain such variability. The time of the last DOAC intake was verified by the medical staff. However, compliance prior to the last intake was based solely on patients’ self-reporting, which could induce a reporting bias.

Basal ACT, measured just before UFH injection, could be used to detect patients outside the therapeutic index with DOACs. Our results indicate that ACT was linearly correlated with basal apixaban and dabigatran concentration but not with basal rivaroxaban concentration. Unfortunately, the study was not powered to determine an optimal threshold for ACT to detect a basal concentration above 300 ng.mL^−1^ or below 30 ng.mL^−1^. Dincq et al. observed similar results in an in vitro study with dabigatran, whereas they observed only a small effect of rivaroxaban or apixaban concentration on ACT [12]. Martin et al. also observed in vitro a good correlation between ACT values and dabigatran concentrations but not for the other two DOACs; at baseline, ACT was longer with dabigatran and shorter with apixaban at similar concentrations [20]. Yamaji et al. observed that the ACT measured before any dose of UFH is higher for dabigatran anticoagulation therapy, suggesting a lower sensitivity of ACT assay for the other DOACs [21].

Given these discrepancies, particularly with respect to the relationship between ACT and rivaroxaban concentration, measuring the specific DOAC concentration just before the procedure could allow clinicians to identify outliers who might benefit from a different anticoagulation protocol.

As expected, the calibrated anti-Xa activity for UFH is influenced by the presence of anti-Xa DOACs as shown by our results before UFH injection with significant activity in patients treated with apixaban and rivaroxaban. The current method for monitoring heparin therapy is based on chromogenic anti-activated factor X assay [22,23] and the specific anti-Xa activity of rivaroxaban and apixaban safely reflected the level of anticoagulation of these two DOACs, respectively [24,25]. However, the simultaneous presence of these molecules interferes with the measurement of the anti-Xa activity in an unpredictable manner, which makes it impossible to monitor the level of anticoagulation with UFH based on this parameter alone [26]. Because dabigatran, through its anti-IIa effect, does not interfere with the anti-Xa assay of UFH, this assay could be used to monitor the anticoagulation level of UFH, although defining the appropriate target may be difficult given the significant variability in the baseline level of anticoagulation by dabigatran.

The initial bolus of UFH resulted in an increase in both ACT and anti-Xa activity in all groups. However, injection of the same weight-adjusted dose in all patients did not result in a similar increase of these parameters. Anti-Xa activity increased significantly more after the initial bolus in patients treated with dabigatran than in those treated with the other two DOACs. In addition, the increase in anti-Xa activity was highly variable among patients, particularly among those treated with rivaroxaban and apixaban. Zeljovic et al. [27] also observed a lower proportion of patients who reached target ACT in the apixaban and rivaroxaban groups as compared to the warfarin and dabigatran groups after the initial bolus of 100 units per kg UFH. Recently, the study by Payne et al. [28] concluded that patients on DOACs require significantly higher doses of heparin to achieve a therapeutic ACT (initial bolus ≥ 150 units per kg) than patients on VKAs, but there was no difference in UFH dose between the different DOACs. However, the study from Payne et al. suffers from the non-standardized protocol for intraprocedural UFH management, with dose and timing of injection left to the discretion of the attending physician.

The currently used target ACT of 300 s was defined in historical studies in patients treated with vitamin K antagonists, which was then extrapolated to patients treated with DOACs; these studies observed a lower incidence of thrombi in left heart chambers with this threshold [13,29,30,31]. In our study, the total dose of UFH administered during the procedure to maintain an ACT > 300 s was significantly higher in patients treated with apixaban than in those treated with rivaroxaban. In addition, the maximum ACT in this group tended to be lower, whereas the maximum anti-Xa activity was significantly higher. Basal ACT was lower in patients treated with apixaban and ACT appears to increase more slowly than anti-Xa activity in this group compared with other DOACs, particularly dabigatran. These results confirm that the interference of apixaban on ACT resulted in the injection of more UFH, leading to a higher anti-Xa activity, and possible a supratherapeutic level of anticoagulation. The only bleeding complication occurred in a patient treated with apixaban who required a high dose of UFH to reach the therapeutic target. This indicates that these interferences may lead to overtreatment or even bleeding complications.

Others studies also concluded that a higher dose of UFH is required to achieve an ACT > 300 s in patients treated with DOAC, with a greater dose in patients treated with rivaroxaban and apixaban as compared to dabigatran [8,10,21,27,32,33,34]. In our study, the increase in ACT during the procedure correlated with the increase in anti-Xa activity of UFH, which is in line with the study from Benali et al., where ACT and anti-Xa activity not only correlated well in patients treated with VKAs, but also with DOACs, although this association was weaker [33]. Thus, ACT appears to be able to appropriately detect heparin boluses, but with a different response among DOACs. Rather than an absolute threshold, targeting a specific increase in ACT may be useful to quantify the effect of UFH during the procedure. However, defining the optimal target increase in ACT to ensure proper anticoagulation is difficult, because the baseline level of anticoagulation by DOAC is highly variable. The meta-analysis of Briceno et al. [29] identified numerous algorithms for ACT-guided heparinization during AF ablation procedure but was not primarily aimed at clarifying the heparinization protocol. Interestingly, in the study of Yamaji et al. [21], the dose of the initial heparin bolus was calculated using the value of ACT before the start of AF ablation, age, sex, and body weight; then, a continuous heparinized infusion was administered to maintain the ACT > 300 s. The authors did not observe any thromboembolic complications and a low incidence rate of bleeding complications. A worldwide survey reported that 78% of 777 centers used ACT-guided administration of heparin to reach 250–300 s, but no report has evaluated the optimal heparinization protocol of the optimal 15-minute ACT [35].

A combined strategy, using categorization of the patient as infra-, supra-, or within the therapeutic range in the basal state as well as a target ACT (absolute or relative), or anti-IIa activity for dabigatran may be interesting but would require further scientific validation before clinical use. Targeting a lower ACT value for patients treated with apixaban may be more appropriate to avoid UFH overdose.

Our study has some limitations, as the small population included in the study did not allow for defining optimal threshold for ACT at the beginning of the procedure. We had to stop inclusions in 2020 because of the huge COVID-19 outbreak in Strasbourg that caused all nonessential procedures, including atrial fibrillation procedures, to be stopped. We had several major waves until mid-2021 that significantly altered our schedule after that. Only five patients were on dabigatran, which significantly reduces the power of analysis performed on these patients. Dabigatran is less frequently used in France, due to its drug interactions (especially amiodarone) and its renal elimination, which can be impaired in case of renal failure frequently found in elderly population [36]. Because this study was only observational, we were unable to perform additional hemostasis analysis. Comparison of ACT and anti-Xa activity results before and after DOAC removal using specific filters could be informative and help define the optimal anticoagulation target [37].

## 5. Conclusions

Today, atrial fibrillation ablation on uninterrupted direct oral anticoagulation is recommended in daily practice. Optimal management of anticoagulation during the procedure is still based on data observed in patients treated with VKAs, and ACT remains the cornerstone of anticoagulant administration.

We report a highly variable baseline anticoagulation status in patients treated with DOAC at the start of the ablation procedure. Without heparin, the ACT appears to reliably reflect apixaban and dabigatran concentrations only. Weight-based bolus heparin variably increases ACT and anti-Xa activity but with differences among DOACs, with the ACT increasing more slowly than anti-Xa activity in patients treated with apixaban. During the procedure, the use of the conventional ACT target of 300 s led to an increase in the dose of heparin administered, particularly in the apixaban group, which could lead to UFH overdose in these patients. Data on dabigatran should be viewed with caution because they are based on only a few patients in our study.

Overall, although based on a small population, our results raise the question of optimal management of intraprocedural heparin therapy and highlight the limitations of ACT testing in these patients. Although ACT monitoring remains the gold standard for these procedures, further studies using specific assays for DOACs before heparin injection and strategy based on different threshold for ACT during the procedure may be of interest.

## Figures and Tables

**Figure 1 jcm-12-02236-f001:**
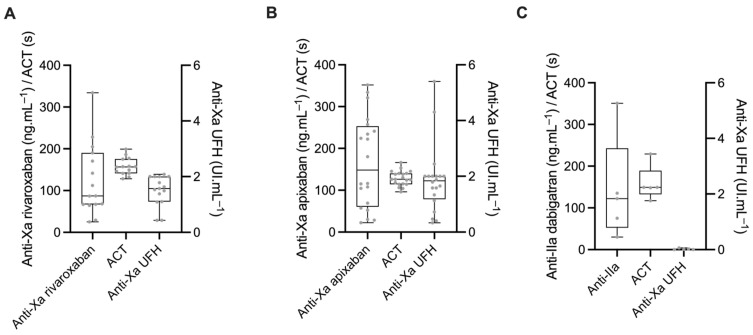
Specific anti-Xa/IIa activity, activated clotting time (ACT), and anti-Xa activity calibrated for unfractionated heparin (UFH) in patients treated with either rivaroxaban (**A**), apixaban (**B**), or dabigatran (**C**).

**Figure 2 jcm-12-02236-f002:**
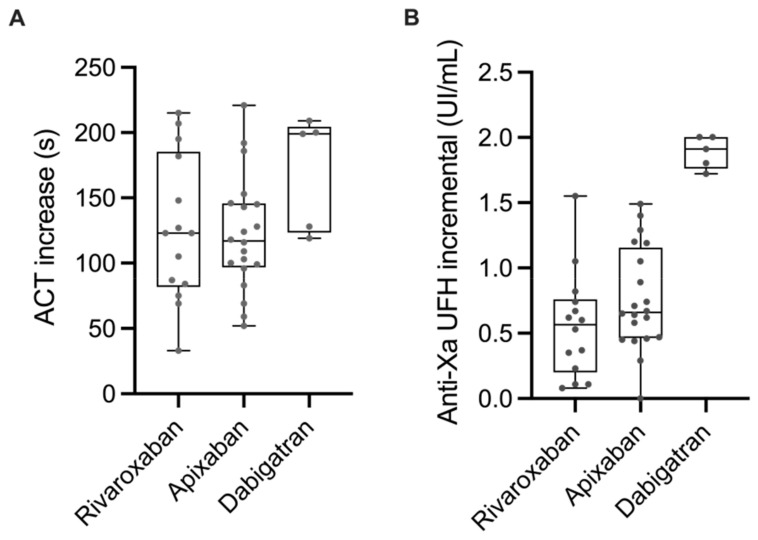
Increase in ACT (**A**) and UFH anti-Xa activity (**B**) after the initial bolus of UFH in patients treated with rivaroxaban, apixaban, or dabigatran.

**Figure 3 jcm-12-02236-f003:**
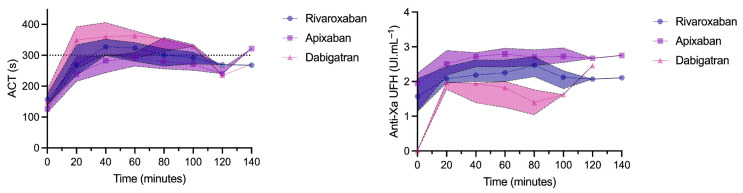
Evolution of activated clotting time (ACT) and unfractionated heparin (UFH) anti-Xa activity during the procedure. Results are expressed as median and interquartile range.

**Table 1 jcm-12-02236-t001:** Patient characteristics, anticoagulant treatment, and type of heart rhythm disorder. Continuous variables are expressed as median [interquartile range] and categorical variables by n (%). BMI = body mass index; eGFR = estimated glomerular filtration rate; DOAC: direct oral anticoagulation.

Characteristics	Variables	Cohort (n = 40)
Demographics	Sex (Male)	22 (55)
	Age (years)	68 (55–71)
	BMI (kg.m^−2^)	27 (24–31)
	CHA2DS2-VASC score	2 (0–3)
	eGFR mL.min^−1^.1.73 m^−2^	75 (62–91)
Type of heart rhythm disorder	Paroxysmal AF	24 (60)
	Persistent/permanent AF	16 (40)
	Left ventricular ejection fraction (%)	65 (56–70)
	Indexed left atrium surface (mL.m^2^)	47 (38–62)
DOAC treatment	Rivaroxaban	15 (37.5)
	Apixaban	20 (50.0)
	Dabigatran	5 (12.5)

## Data Availability

The data presented in this study are available on request from the corresponding author.

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
