# Peer review of "Study of Modifications Induced by Continued Direct Oral Anticoagulant Therapy during Atrial Fibrillation Ablation Procedures on Standard Hemostasis Parameters"

_jcm, 2023, doi:10.3390/jcm12062236_

Round 1
Reviewer 1 Report
The manuscript raises several concerns :
1. Reference missing at line 55, 84
2. Line 162,163 Procedure duration is 105-120 min. The graph in figure 3 shows 100 minutes’ mean data. How do you explain the data and statistical significance?
3. What is the take home message from the ACT and Anti-Xa UFH time point data during the procedure? Do you see a time point dependent correlation between changes in ACT values and in Anti-Xa UFH values?
4. How was the dose of UFH bolus determined? The fact that “dose and timing of injection was left to the discretion of the attending physician” It is unclear on how UFH dose impacts the ACT and Anti-Xa UFH. There is no clear indication for the rationale behind dose selection.
5. Representation of the graphs are confusing. Significance of correlations are not shown clearly in the graphs.
6. Number of data points are too low and variabilities in each experimental state are too high. Therefore it is impossible to draw conclusions based on statistical significance from the provided data.
Author Response
Reviewer 1
The manuscript raises several concerns:
- Reference missing at line 55, 84
We thank Reviewer 1 for this comment. The references have been added.
- Line 162,163 Procedure duration is 105-120 min. The graph in figure 3 shows 100 minutes’ mean data. How do you explain the data and statistical significance?
While some patients underwent a procedure lasting more than 100 minutes, most blood sampling occurred within the first 100 minutes after the start of the procedure. Only very few patients were drawn after 100 minutes, so we felt that adding this data to the graph would not be informative. We modified Figure 3 and added data points after 100 minutes. Based on our small sample size, some differences regarding the procedure duration may occur but they are not statistically significant, as mentioned in the manuscript.
Figure 3. Evolution of activate clotting time (ACT) and unfractionated heparin (UFH) anti-Xa activity during the procedure. Results are expressed in median and interquartile range.
- What is the take home message from the ACT and Anti-Xa UFH time point data during the procedure? Do you see a time point dependent correlation between changes in ACT values and in Anti-Xa UFH values?
One of our questions was to evaluate whether or not ACT correlated with anti-Xa activity during the procedure and whether ACT could detect a bolus of unfractionated heparin. Our data showed that, although imperfect, changes in ACT correlated with changes in anti-Xa. As expected, because of analytical interference between apixaban/rivaroxaban and the specific assay for anti-Xa in UFH, a 100s increase in ACT led to a significantly greater increase in anti-Xa in UFH in patients treated with dabigatran than with apixaban/rivaroxaban. In other words, the level of residual DOAC seems to influence ACT during the procedure with large differences between DOACs, leading to a risk of overdosing/underdosing of patients depending on the anticoagulant used.
This point is discussed in the manuscript, line 259 to 272.
- How was the dose of UFH bolus determined? The fact that “dose and timing of injection was left to the discretion of the attending physician” It is unclear on how UFH dose impacts the ACT and Anti-Xa UFH. There is no clear indication for the rationale behind dose selection.
Current guidelines recommend maintaining an ACT of more than 300 s during the procedure to prevent the occurrence of thromboembolic events. We use a standardized protocol in our hospital with an initial bolus of UFH of 100 IU/kg BW followed by a repeat injection of 50 IU/kg UFH to achieve or maintain our goal. The initial dose was based on the guidelines of HRS/EHRA/ECAS (Calkins et al. Europace 2007). The sentence “dose and timing of injection was left to the discretion of the attending physician” do not refer to our study but to Payne et al. which was indeed a major concern for their study. This has been clarified in the manuscript.
Line 91-93: The first bolus of unfractionated heparin was injected immediately after transseptal puncture (100 IU.kg-1) according to HRS/EHRA/ECAS guidelines
Line 270-272: However, the study from Payne et al. suffers from the non-standardized protocol for intraprocedural UFH management, with dose and timing of injection left to the discretion of the attending physician.
- Representation of the graphs are confusing. Significance of correlations are not shown clearly in the graphs.
We agree with reviewer 1 that there is no graphical representation of the correlation. We were hesitant for a long time to include a graphical representation of the correlation (see below) but since we used Bayesian statistics to which people are not used to, we thought it might be confusing. So we decided to keep only the mention in the manuscript. For example, this is the graphical representation of the association between ACT and the respective anti-Xa activity in patients treated with rivaroxaban (A), apixaban (B) and dabigatran (C). The association is quantified by the posterior probability distribution of the (ß1) coefficient of the linear regression model (ß1 = 0 corresponds to an absence of linear association, the 95% credible interval is the light blue area).
If Reviewer 1 think that this is important to the manuscript, we would be pleased to add in the manuscript.
- Number of data points are too low and variabilities in each experimental state are too high. Therefore it is impossible to draw conclusions based on statistical significance from the provided data.
We agree with reviewer 1 that we have a very small effective population in our study. However, Bayesian statistics are particularly appropriate for working on a small population. As mentioned in the manuscript, several significant differences are observed, and these differences are consistent with the previously published literature. We believe that these results remain interesting as they provide new insights for research on a larger population.
We have moderated our conclusion to include the fact that our results should be taken cautiously because they are obtained from a small population.
Line 339-341: Overall, although based on a small population, our results raise the question of optimal management of intra-procedural heparin therapy and highlight the limitations of ACT testing in these patients.

Reviewer 2 Report
Dear Authors,
I have read submitted paper “Study of modifications induced by continued direct oral anticoagulant therapy during atrial fibrillation ablation procedures on standard hemostasis parameters” with great attention. The Authors aimed to assess the relationship between ACT, drug concentrations and UFH anti-Xa activity in patients treated on uninterrupted DOAC therapy referred for AF ablation. The submitted manuscript is well written and it was a pleasure to read; however, it has some flaws requiring corrections:
1. The main limitation of the study is a relatively small population, combined with uneven distribution of different DOACs used. With only 15, 20 and 5 patients on rivaroxaban, apixaban and dabigatran, respectively - it is really hard to draw any substantial concussions regarding the clinical value of routine intra-procedural ACT testing from this underpopulated study.
2. In the Materials and Methods section (lines 74-75) the Authors wrote: “Patients referred in our center for AF catheter ablation between February 2019 and June 2021 were included”. As there are no inclusion/exclusion criteria given, the reader can assume that all patients undergoing AF ablation in your center were included in the study – this means the total of 40 procedures performed in the 29-month time-span, which gives an average of 16 procedures per year – a really very low number for an AF ablation center. Or maybe only some of the procedures were included in the study – in this case a clear inclusion and exclusion criteria should be provided. This is a major flow and definitely should be clarified.
3. The Section 2.2 Study Endpoints provides the following statement: “For each patient, data collected from the medical record included: age, body mass index (BMI), type and dose of oral anticoagulant, time of last DOAC intake, type of AF, CHA2DS2-VAsc score, Cockcroft-Gault renal function, comedication such as antiplatelet agents, and postoperative complications.” Completing of the study database with the mentioned above data definitely cannot be regarded as the study endpoint, moreover, it is disputable whether a retrospective analysis can have clear endpoints - or just the conclusions, based on the results of collected data analysis.
This critical comments do not affect the value of your work, which I appreciate. They only express the need for general improvement of your manuscript.
Author Response
Please see the attachment
Reviewer 2
I have read submitted paper “Study of modifications induced by continued direct oral anticoagulant therapy during atrial fibrillation ablation procedures on standard hemostasis parameters” with great attention. The Authors aimed to assess the relationship between ACT, drug concentrations and UFH anti-Xa activity in patients treated on uninterrupted DOAC therapy referred for AF ablation. The submitted manuscript is well written and it was a pleasure to read; however, it has some flaws requiring corrections:
- The main limitation of the study is a relatively small population, combined with uneven distribution of different DOACs used. With only 15, 20 and 5 patients on rivaroxaban, apixaban and dabigatran, respectively - it is really hard to draw any substantial concussions regarding the clinical value of routine intra-procedural ACT testing from this underpopulated study.
As respond to reviewer 1, we agree that we have a very small effective population in our study. However, Bayesian statistics are particularly appropriate for working on a small population. As mentioned in the manuscript, several significant differences are observed, and these differences are consistent with the previously published literature. We believe that these results remain interesting as they provide new avenues for research on a larger population.
We have moderated our conclusion to include the fact that our results should not be taken cautiously because they are obtained from a small population. We have also moderated our results on the population of patients treated with dabigatran.
Line 334-338: Data on dabigatran should be viewed with caution because they are based on only a few patients in our study.
Overall, although based on a small population, our results raise the question of optimal management of intra-procedural heparin therapy and highlight the limitations of ACT testing in these patients.
- In the Materials and Methods section (lines 74-75) the Authors wrote: “Patients referred in our center for AF catheter ablation between February 2019 and June 2021 were included”. As there are no inclusion/exclusion criteria given, the reader can assume that all patients undergoing AF ablation in your center were included in the study – this means the total of 40 procedures performed in the 29-month time-span, which gives an average of 16 procedures per year – a really very low number for an AF ablation center. Or maybe only some of the procedures were included in the study – in this case a clear inclusion and exclusion criteria should be provided. This is a major flow and definitely should be clarified.
We completely agree with Reviewer 2 that this may not seem like much.
First, we had to stop inclusions in 2020 because of the huge COVID-19 outbreak in Strasbourg that caused all nonessential procedures, including atrial fibrillation procedures, to be stopped. We had several major waves until mid-2021 that significantly altered our schedule.
The second reason was that we excluded patients who had discontinued DOAC therapy before the procedure when they were instructed to continue. We also excluded patients who were still on VKAs.
Finally, we were able to include only patients scheduled first in the program because our hemostasis laboratory was not able to perform extemporaneous analysis in the afternoon because of personal restrictions.
Taken together, these arguments explain the low number of patients included in this study.
This is now mentioned in the manuscript.
Line 79-82: Patients were included if they continued DOAC therapy until the procedure. They were excluded if they stopped their treatment before the procedure or if they were on VKA. Only patients scheduled first in the program were included to allow extemporaneous hemostasis analysis by the laboratory.
Line 311-314: We had to stop inclusions in 2020 because of the huge COVID-19 outbreak in Strasbourg that caused all nonessential procedures, including atrial fibrillation procedures, to be stopped. We had several major waves until mid-2021 that significantly altered our schedule after that.
- The Section 2.2 Study Endpoints provides the following statement: “For each patient, data collected from the medical record included: age, body mass index (BMI), type and dose of oral anticoagulant, time of last DOAC intake, type of AF, CHA2DS2-VAsc score, Cockcroft-Gault renal function, comedication such as antiplatelet agents, and postoperative complications.” Completing of the study database with the mentioned above data definitely cannot be regarded as the study endpoint, moreover, it is disputable whether a retrospective analysis can have clear endpoints - or just the conclusions, based on the results of collected data analysis. This critical comments do not affect the value of your work, which I appreciate. They only express the need for general improvement of your manuscript.
We fully agree with reviewer 2. We have changed the title of this section to "data collection" to be consistent with the type of our study.

Reviewer 3 Report
Today, atrial fibrillation ablation on uninterrupted direct oral anticoagulation is recommended in daily practice. Optimal management of anticoagulation during the procedure is still based on data observed in patients treated with VKAs, and ACT remains the cornerstone of anticoagulant administration.
The study of the impact of Apixaban, Rivaroxaban and Dabigatran on heparin anticoagulation during atrial fibrillation ablation is a topical one and presents a great interest for medical community.
It is very interesting that in this study, ACT before UFH injection was linearly correlated with Apixaban and Dabigatran concentration, but not with Rivaroxaban concentration and baseline ACT was significantly lower in patient treated with Apixaban.
Another interesting conclusion is that the association between Apixaban and UFH in therapeutic dose (ACT 300 seconds) can induce bleeding complications. Limitations of the study are the retrospective nature of the research and the relatively small number of patients enrolled (40 patients). However, the idea of this study is particularly interesting with possible implications for practical arrhythmology.
Author Response
Reviewer 3
Today, atrial fibrillation ablation on uninterrupted direct oral anticoagulation is recommended in daily practice. Optimal management of anticoagulation during the procedure is still based on data observed in patients treated with VKAs, and ACT remains the cornerstone of anticoagulant administration.
The study of the impact of Apixaban, Rivaroxaban and Dabigatran on heparin anticoagulation during atrial fibrillation ablation is a topical one and presents a great interest for medical community.
It is very interesting that in this study, ACT before UFH injection was linearly correlated with Apixaban and Dabigatran concentration, but not with Rivaroxaban concentration and baseline ACT was significantly lower in patient treated with Apixaban.
Another interesting conclusion is that the association between Apixaban and UFH in therapeutic dose (ACT 300 seconds) can induce bleeding complications. Limitations of the study are the retrospective nature of the research and the relatively small number of patients enrolled (40 patients). However, the idea of this study is particularly interesting with possible implications for practical arrhythmology.
The thank Reviewer 3 for this appreciation.

Reviewer 4 Report
This is very important topic' This study is emphasizing the variability of Heparin effect on ACT in patients treated with DOAC, However, the small number of patients does not allow defining optimal threshold for ACT at the begining of the procedure as stated in the limitation section. Thus, this study does not add information how to deal with heparin in patients treated with different types Of DOAC.
I suggest to increase the number of patients and try to figure out which Heparin regime would work in each DOAC.
- Did you record any thromboembolic complications?
Author Response
Reviewer 4
This is very important topic. This study is emphasizing the variability of Heparin effect on ACT in patients treated with DOAC. However, the small number of patients does not allow defining optimal threshold for ACT at the begining of the procedure as stated in the limitation section. Thus, this study does not add information how to deal with heparin in patients treated with different types Of DOAC.
I suggest to increase the number of patients and try to figure out which Heparin regime would work in each DOAC.
As responded to reviewer 1 and 2, we agree that we have a very small effective population in our study. However, Bayesian statistics are particularly appropriate for working on a small population. As mentioned in the manuscript, several significant differences are observed, and these differences are consistent with the previously published literature. We believe that these results remain interesting as they provide new avenues for research on a larger population.
We have moderated our conclusion to include the fact that our results should be taken cautiously because they are obtained from a small population.
We agree that further research in this area requires larger studies in a larger population. We are working on this, but we would like to perform a more detailed study, including a specific analysis using a DOAC removal system, that would provide a better understanding of hemostasis during the atrial fibrillation procedure. The present study would help us make the case for additional, expensive studies.
We are not sure that with an increased number of analyzed patients, we could find how to deal with heparin in patients treated with different types of DOAC as we could observe a great heterogeneity in baseline pre-procedural level of anticoagulation in each group of DOAC.
Did you record any thromboembolic complications?
We did not record any thromboembolic event in our study. This is now mentioned in the manuscript.
Line 205-206: No thromboembolic event was recorded.

Round 2
Reviewer 4 Report
The authors answered my comments
Author Response
Reviewer 4
The authors answered my comments
We thank the reviewer 4 for his careful reading and comments.
